# Comprehensive Identification of the *Pum* Gene Family and Its Involvement in Kernel Development in Maize

**DOI:** 10.3390/ijms241814036

**Published:** 2023-09-13

**Authors:** Wenqi Feng, Hongwanjun Zhang, Yang Cao, Cheng Yang, Muhammad Hayder Bin Khalid, Qingqing Yang, Wanchen Li, Yingge Wang, Fengling Fu, Haoqiang Yu

**Affiliations:** 1Key Laboratory of Biology and Genetic Improvement of Maize in Southwest Region, Ministry of Agriculture, Maize Research Institute, Sichuan Agricultural University, Chengdu 611130, China; 2National Research Centre of Intercropping, The Islamia University of Bahawalpur, Bahawalpur 63100, Pakistan

**Keywords:** maize, Pumilio RNA-binding proteins, gene expression, kernel development

## Abstract

The Pumilio (Pum) RNA-binding protein family regulates post-transcription and plays crucial roles in stress response and growth. However, little is known about Pum in plants. In this study, a total of 19 *ZmPum* genes were identified and classified into two groups in maize. Although each ZmPum contains the conserved Pum domain, the ZmPum members show diversity in the gene and protein architectures, physicochemical properties, chromosomal location, collinearity, cis-elements, and expression patterns. The typical ZmPum proteins have eight α-helices repeats, except for ZmPum2, 3, 5, 7, and 14, which have fewer α-helices. Moreover, we examined the expression profiles of *ZmPum* genes and found their involvement in kernel development. Except for *ZmPum2*, *ZmPum* genes are expressed in maize embryos, endosperms, or whole seeds. Notably, *ZmPum4*, *7*, and *13* exhibited dramatically high expression levels during seed development. The study not only contributes valuable information for further validating the functions of *ZmPum* genes but also provides insights for improvement and enhancing maize yield.

## 1. Introduction

Post-transcriptional regulation of gene expression employs a wide range of RNA-binding proteins (RBPs) and plays crucial roles in finely controlling protein synthesis in a spatial and temporal manner. RBPs contribute to regulating RNA processing, mRNA transport, stability, and translation by targeting specific 3′-untranslated regions (UTRs) of target mRNA [1,2,3]. Additionally, RBPs can collaborate with ribosomal protein binding sites within the 5′-UTR or microRNAs (miRNAs) to regulate mRNA metabolism [4,5]. The Pumilio (Pum) RNA-binding proteins, known as Puf proteins, are a kind of RBP and are well characterized in animals and fungi [6,7], but little is known about the Pum family in plants.

The Pum proteins exhibit high conservation of the Pumilio Homology Domain (Pum-HD) in various organisms [8,9,10]. The Pum-HD possesses a unique crescent-shaped structure and is necessary for RNA binding [11,12,13]. Typically, this domain consists of imperfect tandem Puf repeats forming as α-helices each containing approximately 36 amino acids, and allows Pum proteins specifically to interact with mRNA to regulate post-transcription processes [11,14,15,16]. Within each Puf repeat, the second α-helix serves as the primary binding interface between the Pum and the target RNA [17]. The Puf repeat binds to a single RNA base through hydrogen bonds, van der Waals interactions, and base stacking. The binding of Pum and RNA can be facilitated by three conserved amino acid side chains within each repeat, which allows Pum proteins to selectively bind to specific mRNA sequences [8]. Generally, Pum proteins are recognized by the conserved UGUA core motif situated within the 3′-UTR of target mRNA [18,19]. Interestingly, it was also found that Pum proteins interact with other proteins to inhibit translation or the initiation of mRNA decay processes [17]. The human Pum interacts with NORAD (non-coding RNA activated by DNA damage) to preserve genomic stability [20]. The PUF5p forms a complex with Pop2p, a component of the deadenylase complex, to regulate mRNA decay in yeast [21]. Specifically, Pum proteins stimulate deadenylation and decapping to accelerate mRNA turnover and reduce translation efficiency [22].

Although Pum proteins show high conservation in the Pum-HD sequence, there is a high diversity of *Pum* gene members in plants. For example, there are 2, 2, 2, 6, 10, and 11 Pum genes in the genome of *Drosophila melanogaster*, human, mouse, *Saccharomyces cerevisiae*, *Trypanosoma cruzi*, and *Caenorhabditis elegans*, respectively [8,11,23,24]. On the contrary, there are 31, 26, 28, 26, 22, and 20 *Pum* genes in the genome of *Arabidopsis lyrata*, *Arabidopsis thaliana*, *Malus domestica*, *Glycine max*, *Oryza sativa* ssp. *Indica*, and *Oryza sativa* ssp. *Japonica*, respectively [17,24,25]. This suggests that Pum proteins are involved in a wide range of post-transcriptional/translational regulations to control growth and development and cope with environmental stresses in plants. However, little is known about the roles of Pum genes in plants. In *Arabidopsis*, AtPum5 is involved in cucumber mosaic virus (CMV) and abiotic stress response and negatively regulates salt and drought tolerance by binding to 3′-UTR of the abiotic stress-responsive genes containing the Pum RNA-binding motifs at the 3′-UTR [26,27]. AtPum9 binds to target transcripts to trigger mRNA degradation via Pum-HD at the C-terminal and interacts with DCP2 (the catalytic subunit of the decapping complex) to positively regulate heat stress and seed dormancy mediated by REDUCED DORMANCY5 encoding a PP2C phosphatase [28,29]. AtPum23, a nuclear-localized protein, is required for normal plant growth including leaf development and organ polarity, as well as being involved in salt response mediated by ABA signaling via regulating rRNA processing [30,31,32]. AtPum24 is an atypical Pum protein and reduces mRNA stability of the BTB/POZMATH (BPM) gene family by directly binding to their 3′-UTR to regulate plant development, seed maturation, and starch, protein, and oil biosynthesis [33,34]. To date, the *Pum* gene family has only been identified genome-wide in *Arabidopsis* and rice [17]. In addition, the roles of Pum genes in plants are largely unknown.

Maize is one of the most important crops and plays a crucial role in ensuring food and economic security [35]. The primary components including starch, protein, and oil are stored within maize seeds, which account for approximately 90% of the total dry seed weight. The content and composition of these components in maize kernels have a significant impact on their quality [36]. Hence, the maize kernel is a valuable resource for human consumption, animal feed, and bioenergy applications. In this study, we focused on comprehensively exploring the maize *Pum* gene family. We identified 19 *ZmPum* genes and investigated their physicochemical properties, phylogenetic relationships, chromosome localization, gene and protein-conserved domain structure, cis-acting elements, and expression patterns in kernel development. The main objective is to provide valuable insights into the underlying role of *ZmPum* genes in regulating seed development in maize and contribute to the analysis of the Pum family in plants.

## 2. Results

### 2.1. The ZmPum Family in Maize

In the maize genome, a total of 19 candidate genes encoding Pum proteins were identified and defined as *ZmPum1* to *ZmPum19*. The coding sequence of the *ZmPum* gene was 1191 to 3009 bp in length, encoding 396 to 1002 amino acids (aa), with a molecular weight (MW) of 42.49 to 109.02 kDa. The isoelectric points (PIs) of ZmPum1, 2, 3, 5, 7, 13, 14, and 18 proteins were more than 7.00, the other eleven ZmPum proteins had PIs ranging from 5.70 to 6.73. The instable indices of ZmPum1, 6, 17, 18, and 19 were less than 40.00, while others were more than 40.00. All ZmPum proteins were hydrophilic proteins with a grand average of hydropathicity (GRAVY) < 0. Fourteen ZmPum proteins were predicted to show nuclear localization, only ZmPum1, 2, and 13 showed cytoplasm localization, ZmPum6 showed vacuole localization, and ZmPum18 showed chloroplast localization (Table 1). The diversity of properties of ZmPum proteins may imply their different roles.

The phylogenetic analysis showed that maize ZmPum members were clustered into four subclades within the phylogenetic tree based on the similarity of their amino acid sequences with AtPum proteins. ZmPum3, 4, 6, 8, 9, 11, 12, 13, 15, and 19 were grouped in subclade I. ZmPum1, 10, 16, 17, and 18 were clustered into clade II. Two (ZmPum7 and 14) and three (ZmPum2, 5, and 24) of them were grouped in clades III and IV, respectively (Figure 1).

### 2.2. Protein Architectures of ZmPum

As shown in Figure 2, six conserved motifs were discovered in ZmPum proteins and named motifs 1–6. The majority of ZmPum proteins contained motifs 1, 2, and 3, excluding ZmPum3 and ZmPum5. Among them, motifs 1, 2, 3, 4, and 5 contribute to the composition of Pum domains. Conserved domain analysis showed that the ZmPum proteins could be divided into two groups: typical Pum and atypical Pum. ZmPum3, 5, 7, and 14 belonged to one subgroup and were atypical Pum families because they possessed few Pum domains. The other 14 ZmPum members were grouped into another clade and were typical Pum proteins.

Three-dimensional structures of 19 ZmPum proteins were predicated using structure-based analysis in the Ensembl database. All ZmPum proteins contained the conserved Pum domain with a different number of α-helix repeats. The ZmPum proteins from the same subgroup exhibited a similar three-dimensional structure. However, the members from subclade I were atypical Pum proteins containing fewer imperfect Pum domains. The ZmPum members from group II, except ZmPum16, had eight Pum domains at the C-terminal region (Figure 2 and Figure 3).

### 2.3. Chromosomal Location and Gene Duplication of ZmPum

The information on chromosome location of *ZmPum* genes was obtained from the maizeGDB database and used for visualization of 19 *ZmPum* mapping to the maize genome (Figure 4). In detail, there was no *ZmPum* gene on chromosome 3. Other *ZmPum* genes were unevenly distributed on the other 9 maize chromosomes. There were five, three, three, two, and two *ZmPum* genes on chromosomes 1 (*ZmPum1*, *2*, *3*, *4*, and *5*), 4 (*ZmPum8*, *9*, and *10*), 10 (*ZmPum17*, *18*, and *19*), 2 (*ZmPum6* and *7*), and 7 (*ZmPum13* and *14*), respectively. The *ZmPum11*, *12*, *15*, and *16* genes were mapped on chromosomes 5, 6, 8, and 9, respectively. The results of gene duplication analysis showed that seven segmental duplication events were detected among 19 *ZmPum* genes, and each gene pair was located on a distinct chromosome, including pairs of *ZmPum1* and *16*, *ZmPum2* and *16*, *ZmPum4* and *8*, *ZmPum6* and *19*, *ZmPum7* and *14*, *ZmPum9* and *11*, as well as *ZmPum12* and *15* (Figure 4).

Additionally, the synteny between the *Pum* gene families in the maize and rice, as well as maize and *Arabidopsis* genomes, was also examined. It was revealed that there were nineteen pairs of *Pum* orthologous genes in maize and rice, and five *Pum* gene pairs in maize and *Arabidopsis* (Figure 5; Appendix A).

### 2.4. Gene Structure and Cis-Elements of ZmPum

To further examine the organization of the exons and introns of *ZmPum* genes, the CDS and the corresponding gDNA sequence of each *ZmPum* gene were analyzed using GSDS. It showed that the numbers of exons and introns varied greatly among different ZmPum members (Figure 6), which ranged from 5 to 10 exons unevenly. For example, *ZmPum3*, *5*, *9*, and *11* had 10 exons. *ZmPum6*, *12*, and *15* had 9 exons, *ZmPum4* and *19* both possessed 8 exons, and the other *ZmPum* genes had fewer exons.

Cis-elements analysis showed that abundant elements involved in hormone response were identified in promoter sequences of *ZmPum* genes, such as ABRE, AuxRRcore, P- box/GARE, TCA, and TGACG/CGTCA motif elements, which were responsive to ABA, auxin, gibberellin, SA, and MeJA, respectively (Figure 7; Appendix A). In general, 13 *ZmPum* genes (68.4%) possessed auxin-responsive elements (AuxRRcore). Thirteen *ZmPum* genes had gibberellin-responsive elements (P-box or GARE). Moreover, *ZmPum12* carried RY-elements involved in seed-specific regulation. Meanwhile, MBS, MBSI, and LTR elements involved in drought, flavonoid biosynthetic regulation, and low-temperature response, respectively, were found in their promoters. In total, 11 *ZmPum* genes had MBS elements, 10 *ZmPum* genes (52.6%) had LTR elements, and 2 *ZmPum* genes had MBSI elements.

### 2.5. Tissue-Specific Expression Patterns of ZmPum

The results of tissue-specific expression analysis showed that the *ZmPum* genes were classified into three groups in terms of different expression patterns (Figure 8). *ZmPum3*, *4*, *5*, *9*, *11*, *15*, and *18* could be clustered into one group and highly expressed in pollinated internodes, embryos, endosperm, and whole seeds. *ZmPum1*, *2*, *6*, *7*, *8*, *17*, and *19* were clustered into one group and expressed in bicellular male gametophytes, microspores, and sperm cells. Interestingly, *ZmPum6* and *17* were also slightly expressed during seed development. In addition, *ZmPum10*, *12*, *13*, *14*, and *16* were grouped into one branch and dominantly expressed in endosperm and whole seeds. Meanwhile, *ZmPum12* and *14* exhibit high expression in pollinated internodes and slight expression in seminal, silk, roots, and bicellular male gametophytes. The results suggest that the *ZmPum* genes may play essential roles in regulating maize growth and development, particularly in the formation of seeds.

### 2.6. ZmPum Regulates Kernel Development

High-resolution transcriptome data in maize endosperms ranging from 48 to 144 h (En48-144) after pollination (HAP) with a time interval of 24 h were recently reported by Fu et al. [37]. Interestingly, except for the *ZmPum2* gene, the expression of the other 18 *ZmPum* genes was detected during En48-144 HAP (Figure 9). *ZmPum1*, *4*, *8*, *9*, *11*, *16*, *17*, and *18* exhibited a high expression level in En48 HAP and a low transcript level in En72, En96, En120, and En144 HAP. While *ZmPum3*, *5*, *6*, *7*, *10*, *12*, *13*, *14*, *15*, and *19* showed high expression levels in En72, En96, En120, and En144 HAP, but low expression in En48 HAP.

Subsequently, qRT-PCR was performed and used to confirm the expression of *ZmPum* genes in maize kernel development at 15, 20, and 25 days after pollination (DAP). As shown in Figure 10, the expression of *ZmPum2* was not detected in any samples. However, the *ZmPum4*, *7*, and *13* genes showed extremely high expression levels in the kernel of 15, 20, and 25 DAP. The *ZmPum3*, *8*, *9*, *10*, *11*, *12*, *14*, *16*, and *17* genes exhibited a high transcript level. Inversely, the expression of the *ZmPum1*, *5*, *6*, *18*, and *19* genes in the kernel of 15, 20, and 25 DAP was lower than others.

The above results suggest that the *ZmPum* genes play a key role in kernel development and may have distinct roles and functions during endosperm development.

## 3. Discussion

To date, Pum proteins have been identified as a kind of RBP to regulate gene post-transcription via conserved Pum-HD [11,13,14,29]. In the present study, a total of 19 ZmPum members were identified in the maize genome (Table 1; Figure 1). Meanwhile, ZmPum were grouped into typical and atypical Pum with different numbers of Pum domains and showed the diversity of gene structures (Figure 2, Figure 3 and Figure 6), which was similar to the AtPum family [17,30]. However, the number of Pums in different organisms is variable and showed higher diversity in plants, which could be explained by whole-genome duplications [17]. It was also found that there were some paralogous and orthologous *Pum* gene pairs in the maize genome and between the rice and *Arabidopsis* genomes (Figure 4 and Figure 5). Tandem duplication and segmental duplication have played essential roles in expanding gene families during the species’ evolutionary history [38,39].

Plants evolved various mechanisms, including physiological, biochemical, and molecular changes, for survival under adverse conditions [40]. Herein, improvement of crop performance and yield under environmental stimuli is a crucial goal during sustainable agriculture production to ensure food security. Although it is not well known for *Pum* in plants, few available reports show that *Pum* can respond to stress such as heat, drought, salt, osmotic, ABA, dark, light, brassinolide, and glucose, as well as regulate development [26,27,28,32,34,41]. In *Arabidopsis*, AtPum1 to 6 are specifically associated with genes related to shoot stem cell maintenance genes [25]. *AtPum5* regulates CMV infection and salt tolerance [26,27,42]. Similarly, rice *Pum* genes respond to biotic and abiotic stress including *Magnaporthe oryzae* and *Nilaparvata lugens* infections, cold, drought, auxin, and cytokinin [30]. However, the *atpum23* mutant exhibited delayed germination rates compared to wild-type plants [32]. Reducing *AtPum24* expression resulted in abnormal seed maturation, wrinkled seeds, and lower seed oil contents, but higher starch and sugar contents. Inversely, overexpression of *AtPum24* increased seed fatty acid, size, and weight [33]. The findings suggest that *Pum* regulates seed development in plants.

In maize, the endosperm is the main nutritive tissue and accounts for approximately 90% of the total dry seed weight. Starch, protein, and oil are the primary storage components within maize endosperm [43]. The improvement of kernel traits holds significant importance for cultivating new maize germplasm with superior quality and high yield. Statistical analysis of transcriptome data from MaizeGDB showed that 84% (16/19) of *ZmPum* genes are highly expressed in seeds after pollination, except *ZmPum2*, *8*, and *19* (Figure 8). In maize, the early endosperm development phase plays a key role in kernel development and comes to an end at 144 h HAP. Afterward, the endosperm shifts to rapid cell proliferation and differentiation, and enters the filling stage [38,44,45,46]. Here, it was also found that 18 *ZmPum* genes exhibited high transcription activity in En48, En72, En96, En120, or En144 HAP (Figure 9). Most *ZmPum* genes maintained high expression levels in maize seeds (Figure 10). These results indicate that the *ZmPum* family is involved in the regulation of early endosperm development.

Insights into the molecular interactions between Pum proteins and RNA bases have been well revealed in some eukaryotes but are still urgently needed to be explored in plants. Plants possess highly complex genomes with a high number of Pum members, which implies that Pum has specific target mRNA and function in plants [25]. In summary, we identified 19 *ZmPum* genes and found their involvement in kernel development in maize. In a further study, the function and molecular mechanism of ZmPum genes in regulating seed traits will be revealed. Overall, the study provides a valuable reference to improve crops through genetic engineering approaches.

## 4. Materials and Methods

### 4.1. Identification of ZmPum and Phylogenetic Analysis

To identify the ZmPum in maize, the database of amino acid of Zm-B73 V5.0 and the AtPum protein sequences were retrieved from MaizeGDB (https://maizegdb.org/, accessed on 5 April 2023) and TAIR (https://www.arabidopsis.org/accessed on 5 April 2023), respectively. The local BLASTP was conducted for ZmPum searching using the AtPum sequences as a reference. Additionally, the hidden Markov model (HMM) files of Pum-HD were acquired from the Pfam database (http://pfam.xfam.org/, accessed on 7 April 2023) and used to search for ZmPUM protein sequences [47]. The properties of the ZmPUM proteins such as molecular weight, isoelectric point, hydrophilic index, stability coefficient, and grand average of hydropathicity (GRAVY) were analyzed using the ExPASy tool (www.expasy.org/tools/, accessed on 7 April 2023) [48]. The subcellular localization was predicted using the BUSCA tool (http://busca.biocomp.unibo.it/, accessed on 7 April 2023) [49]. For the phylogenetic analysis, the amino acid sequences of Pum from maize and Arabidopsis were aligned and used for constructing a phylogenetic tree by Mega 7.0, employing 1000 bootstrap replicates to assess the reliability of the tree topology.

### 4.2. Conserved Motif, Domain, and Structures Analysis

The conserved motifs of ZmPum proteins were identified using the MEME online program (http://meme.sdsc.edu/meme/intro.html, accessed on 5 April 2023) [50]. To verify the conserved Pum-HD of ZmPum proteins, the protein sequences of ZmPum were analyzed using an online tool in the SMART database (https://smart.embl.de, accessed on 12 April 2023) and NCBI-CDD database (https://www.ncbi.nlm.nih.gov/Structure/cdd/wrpsb.cgi, accessed on 14 April 2023) [51,52]. Meanwhile, the 3D structures of ZmPum proteins were predicted by the Swiss-Model tool (https://swissmodel.expasy.org/interactive/, accessed on 14 April 2023) [53]. Subsequently, the quality of the predicted protein structures was evaluated using the SAVES server (http://nihserver.mbi.ucla.edu/SAVES/, accessed on 15 April 2023) [54].

### 4.3. Gene Structure and Duplication Analysis

For the examination of exon/intron structures, the coding sequences (CDS) and genomic DNA (gDNA) sequences of ZmPum genes were analyzed using the Gene Structure Display Server 2.0 (GSDS) (http://gsds.cbi.pku.edu.cn/, accessed on 18 April 2023) [55]. Chromosome localization of the *ZmPum* gene was obtained from maizeGDB and visualized using TBtools [56]. Gene duplications of the *ZmPum* genes were identified and examined by TBtools. The collinearity of the orthologous *Pum* genes between maize, rice, and *Arabidopsis* was determined and plotted using MCScanX Circos within TBtools, respectively.

### 4.4. Cis-Acting Elements and Expression Analysis of ZmPum

To determine the cis-elements in *ZmPum* gene promoters, the 2000 bp promoter sequences of *ZmPum* genes were acquired from the maizeGDB and analyzed using the PlantCARE software (http://bioinformatics.psb.ugent.be/webtools/plantcare/html/, accessed on 20 April 2023) [57]. Furthermore, the TBtools software was employed to visualize the composition of cis-elements in promoters.

To examine the specific expression patterns of *ZmPum* genes in maize, expression data for maize’s different developmental stages and tissues and high-resolution transcriptome data from time points ranging from 48 to 144 h after pollination (HAP) were obtained from qTeller in MaizeGDB (https://qteller.maizegdb.org/, accessed on 22 April 2023) and Fu et al. [37], respectively. Then, the expression of *ZmPum* genes was analyzed and used to create a heatmap using TBtools.

### 4.5. RNA Extraction and Quantitative Real-Time PCR (qRT-PCR)

The maize seeds were sampled at 15, 20, and 25 DAP and used to extract total RNA using RNAiso plus kit (TaKaRa, Dalian, China) according to the manufacturer’s instructions. The concentration and purity of the RNA samples were determined using NanoDrop^TM^ OneC (ThermoScientific, Waltham, USA). The qualified RNA samples were then reverse-transcribed into cDNA using the PrimeScript^TM^ reagent kit (TaKaRa, Dalian, China) and used for qRT-PCR. The qRT-PCR was conducted using the TransScript^®^ II Two-Step RT-PCR SuperMix (Transgen, Beijing, China) in the Bio-Rad CFX96TM Real-Time PCR system. The specific primers of each *ZmPum* gene were designed using Primer-BLAST (https://www.ncbi.nlm.nih.gov/tools/primer-blast/index.cgi?LINK_LOC=BlastHome, accessed on 25 May 2023), synthesized at Sangon Biotech (Chengdu, China), and listed in Appendix A. In addition, a 139 bp fragment of maize *ZmTUB* gene was amplified using primers T-F/T-R (Appendix A) and used as an internal reference for normalization. To determine the relative expression levels of the *ZmPum* genes, the 2^−ΔΔCT^ method was employed [58]. This assay was conducted with three biological and technological replicates.

## 5. Conclusions

In summary, we identified 19 *ZmPum* genes and found their involvement in kernel development in maize. In the next study, the function and molecular mechanism of *ZmPum* genes in regulating seed traits will be revealed. Overall, the study provides a valuable reference to improve crops through genetic engineering approaches.

## Figures and Tables

**Figure 1 ijms-24-14036-f001:**
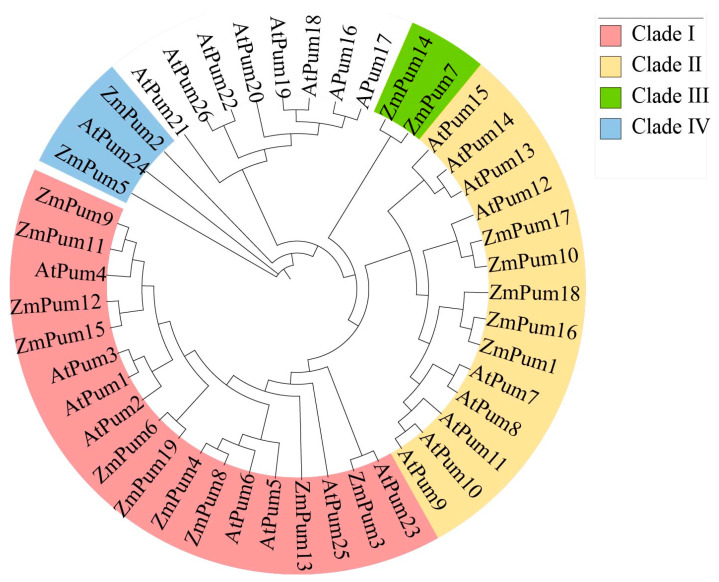
Phylogenetic tree of ZmPum and AtPum proteins. The phylogenetic tree was constructed using MEGA7 software. The full-length amino acid sequences of 19 ZmPum and 26 AtPum proteins were aligned and used to construct a tree with 1000 bootstrap replicates to support the branching patterns.

**Figure 2 ijms-24-14036-f002:**
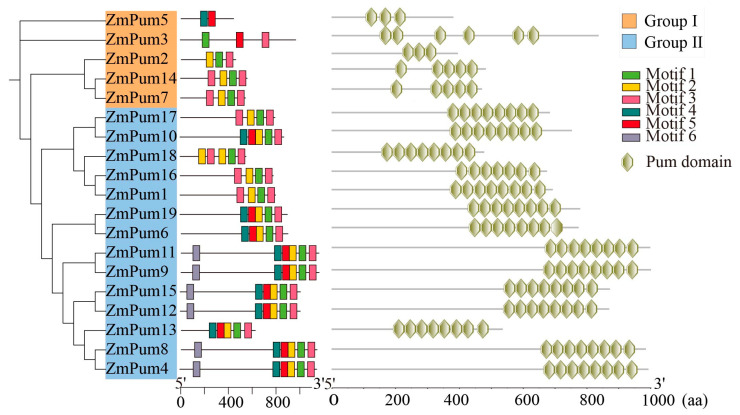
Diagram of motifs and domains of the ZmPum proteins.

**Figure 3 ijms-24-14036-f003:**
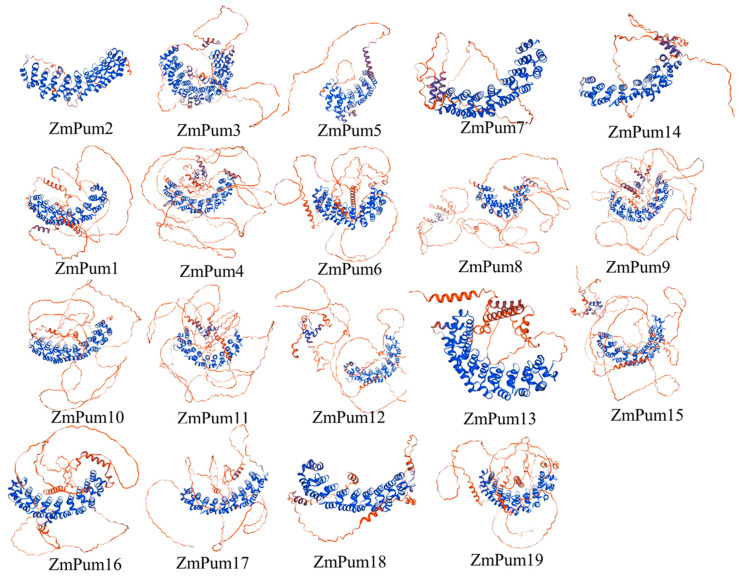
The structure modeling of ZmPum proteins.

**Figure 4 ijms-24-14036-f004:**
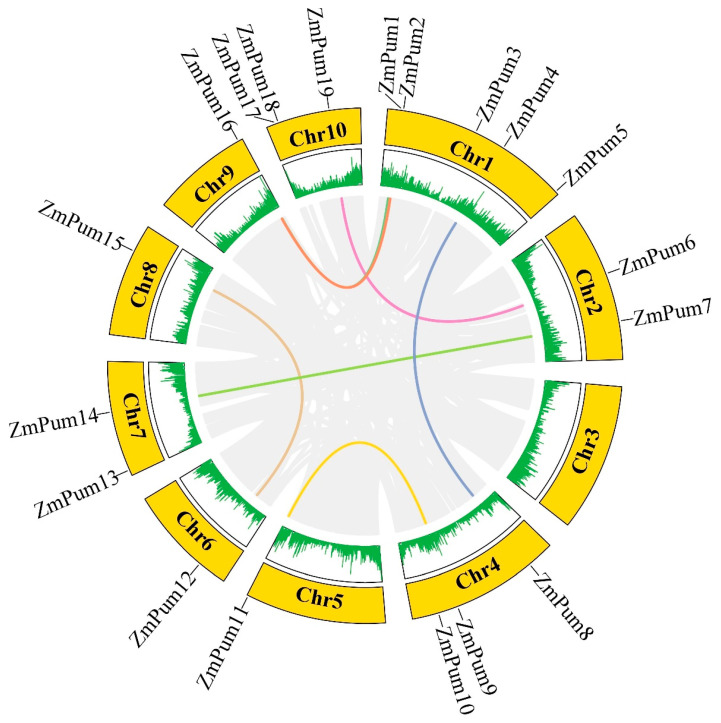
The location and segmental duplication of ZmPum members. The outer circle with chromosome numbers illustrates different maize chromosomes. The maize gene density is displayed by short-green lines in the inner circle. The gray lines indicate all segmental duplications in the maize genome, and the colored lines indicate segmentally duplicated *ZmPum* gene pairs.

**Figure 5 ijms-24-14036-f005:**
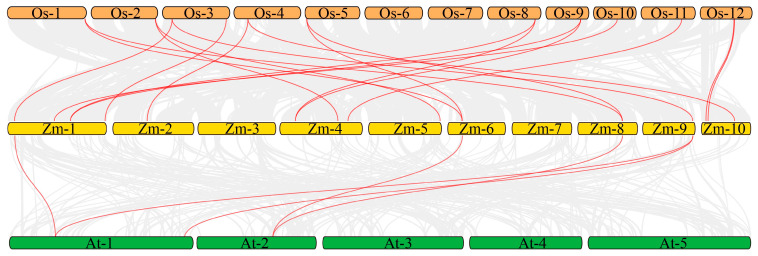
Synteny analysis of *Pum* genes. Gray lines: all collinear blocks within maize, rice, and *Arabidopsis* genomes. Red lines: the synteny of *Pum* gene pairs. The species names with the prefixes Zm, Os, and At indicate maize, rice, and *Arabidopsis*, respectively.

**Figure 6 ijms-24-14036-f006:**
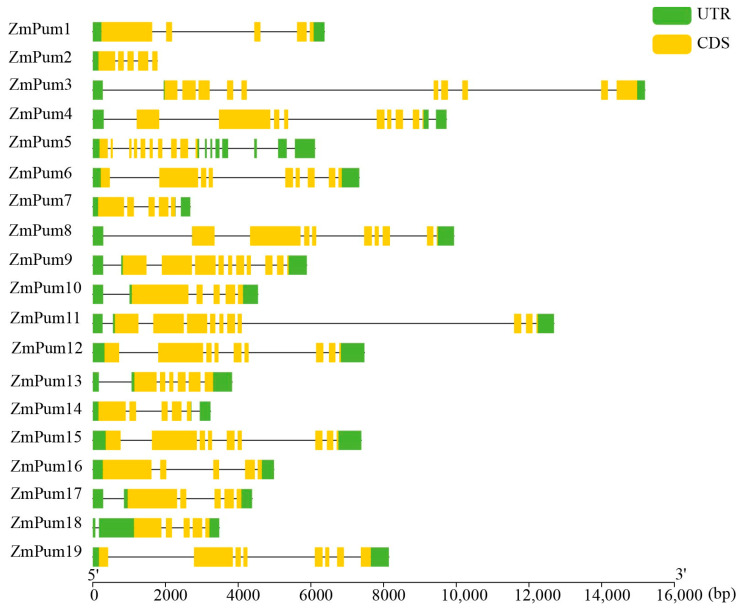
Exon and intron composition of *ZmPum* genes. The CDS and gDNA sequences of *ZmPum* genes were retrieved from maizeGDB and analyzed using the Gene Structure Display Server 2.0 (GSDS).

**Figure 7 ijms-24-14036-f007:**
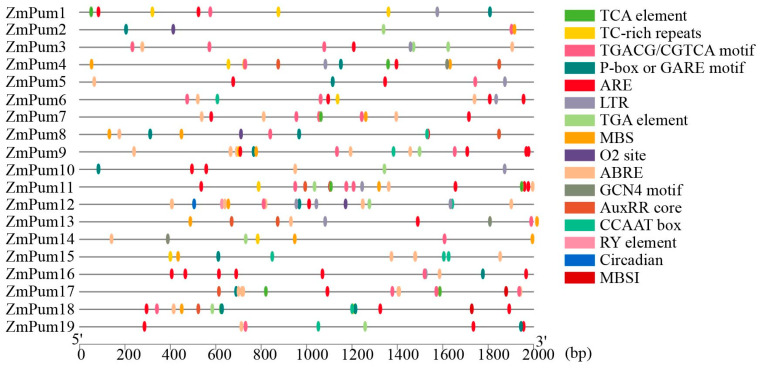
Predicted cis-regulatory elements in *ZmPum* promoters. The promoter sequence of each *ZmPum* gene was analyzed by PlantCARE.

**Figure 8 ijms-24-14036-f008:**
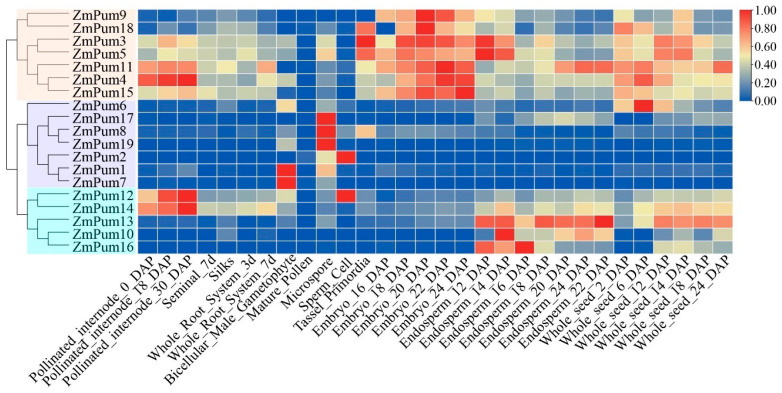
Expression profiles of the *ZmPum* genes in different tissues. The colored scale represents expression data.

**Figure 9 ijms-24-14036-f009:**
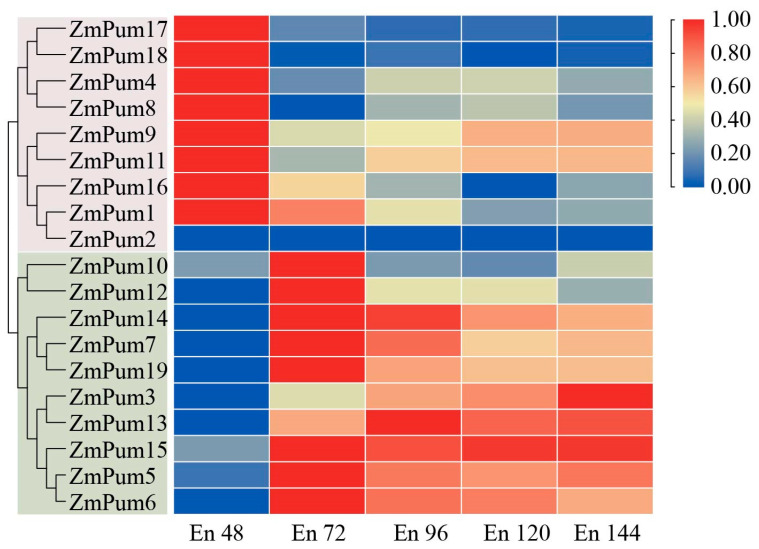
Heatmap of *ZmPum* expression after pollination in the endosperm. En48, En72, En96, En120, and En144 indicate the endosperm at 48, 72, 96, 120, and 144 h after pollination, respectively.

**Figure 10 ijms-24-14036-f010:**
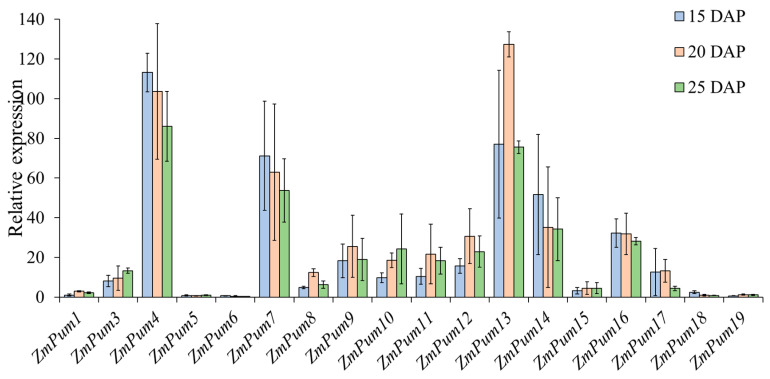
Relative expression level of *ZmPum* genes in maize kernel development. Here, 15, 20, and 25 DAP indicate the seed at 15, 20, and 25 days after pollination, respectively.

**Table 1 ijms-24-14036-t001:** The ZmPum members in maize.

Gene Name	ID	Length (bp) ^a^	Length (aa) ^b^	MW (kDa)	PI	II	GRAVY	Local ^c^
*ZmPum1*	Zm00001eb006950	2085	694	76.22	8.12	39.11	−0.219	C
*ZmPum2*	Zm00001eb008320	1191	396	44.95	8.04	53.99	−0.002	C
*ZmPum3*	Zm00001eb028000	2517	838	92.82	9.32	42.50	−0.548	N
*ZmPum4*	Zm00001eb035700	2985	994	108.97	6.02	47.08	−0.470	N
*ZmPum5*	Zm00001eb063600	1149	382	42.49	9.37	42.96	−0.245	N
*ZmPum6*	Zm00001eb087930	2328	775	86.48	5.70	39.96	−0.420	V
*ZmPum7*	Zm00001eb098710	1419	472	53.14	7.63	66.34	−0.202	N
*ZmPum8*	Zm00001eb174630	2961	986	108.05	6.00	49.57	−0.476	N
*ZmPum9*	Zm00001eb191290	3009	1002	109.02	6.16	50.92	−0.450	N
*ZmPum10*	Zm00001eb200460	2265	754	83.93	6.43	40.51	−0.479	N
*ZmPum11*	Zm00001eb259050	3003	1000	108.94	6.36	50.17	−0.459	N
*ZmPum12*	Zm00001eb266980	2616	871	94.92	6.12	54.14	−0.381	N
*ZmPum13*	Zm00001eb301130	1614	537	58.77	8.58	49.65	−0.165	C
*ZmPum14*	Zm00001eb311340	1443	480	54.23	8.16	62.40	−0.248	N
*ZmPum15*	Zm00001eb355970	2622	873	95.18	6.08	52.47	−0.390	N
*ZmPum16*	Zm00001eb401310	2031	676	74.14	6.73	46.54	−0.182	N
*ZmPum17*	Zm00001eb408690	2058	685	76.04	6.15	36.44	−0.336	N
*ZmPum18*	Zm00001eb409750	1440	479	52.63	8.33	31.04	−0.037	Ch
*ZmPum19*	Zm00001eb419690	2343	780	87.27	5.92	39.25	−0.403	N

^a^ The CDS length of each *ZmPum* gene. ^b^ The amino acid length of each ZmPum protein. ^c^ Subcellular localization of each ZmPum protein. MW: molecular weight. PI: isoelectric point. II: instable index. GRAVY: grand average of hydropathicity. C: cytoplast. N: nuclear. V: vacuole. Ch: chloroplast.

## Data Availability

All data are included in the article and Appendix A.

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
