# Peer review of "Comprehensive Identification of the Pum Gene Family and Its Involvement in Kernel Development in Maize"

_ijms, 2023, doi:10.3390/ijms241814036_

Round 1
Reviewer 1 Report
I congratulate the authors for their work and manuscript. It is well written and deals with the complex family of Pum gene family and their involvement in kernel development in maize. The in silico identification and characterization of the 19 family members in the genome of Maize was well done and expression data allowed for the beginning of understanding their roles. I believe the work will be interesting to readers of IJMS and will recommend acceptance in its current form.
Please review the proof carefully as there were multiple instances of words breaking due to different linebreaks during the preparation of the draft.
Reviewer 2 Report
The first paragraph of the introduction should clearly explain the scope, importance, and incentive of the work in a manner that is understandable by the readers. Add recent relevant studies and clearly write the main objectives at the end of the introduction.
Apart from arabidopsis, I suggest adding a synteny relationship with closely related species of maize.
I suggest performing the subcellular localization analysis and also
validation by BIFC and yeast one-hybrid (Y1H) assays. It gives more fruitful information.
Add the statistical details in Figure 10, better use the uniform term DAP in the entire manuscript.
Minor editing of English language required
Reviewer 3 Report
The manuscript "Comprehensive identification of the Pum gene family and their involvement in kernel development in maize" aims to explore and investigate the physicochemical properties, phylogenetic relationships, chromosome localization gene, and protein-conserved domain structure of ZmPum gene family throughout the kernel development. The manuscript is well-written and technically sound. The results support the conclusion.
I have minor revisions for the authors to improve the quality of the manuscript.
Punctuation is necessary in line 34.
There is no reference in the first paragraph of the Introduction.
Line 35: The authors mention that "many well-studied RBPs have been found" but they don't use any reference right after the phrase. You will need at least more than 5 citations to give the reader the opportunity to know more about these "many RBPs".
Line 237: Please remove the hyphen in "number".
Same for lines 239, 240, 241, 261, 278 and so on. Please revise the whole manuscript.
Line 254: Species = italics.
Round 2
Reviewer 2 Report
I recommend the manuscript for publication
Minor editing of english language required